# EmotionTalk: An Interactive Chinese Multi-modal Emotion Dataset With Rich Annotations

## Abstract

In recent years, emotion recognition has played an increasingly crucial role in applications such as human-computer interaction, mental health monitoring, and sentiment analysis. Although a large number of sentiment analysis datasets have emerged for mainstream languages such as English, high-quality and naturally recorded multimodal dialogue datasets remain extremely scarce for Chinese, given its unique linguistic characteristics, rich cultural connotations, and complex multimodal interaction features. In this work, we propose **EmotionTalk**, an interactive Chinese multimodal emotion dataset with rich annotations. This dataset provides multimodal information from 19 actors participating in dyadic conversational settings, incorporating acoustic, visual, and textual modalities. It includes 23.6 hours of speech (19,250 utterances), annotations for 7 utterance-level emotion categories (happy, surprise, sad, disgust, anger, fear, and neutral), 5-dimensional sentiment labels (negative, weakly negative, neutral, weakly positive, and positive) and 4-dimensional speech captions (speaker, speaking style, emotion and overall). The dataset is well-suited for research on unimodal and multimodal emotion recognition, missing modality challenges, and speech captioning tasks. To our knowledge, it represents the first high-quality and versatile Chinese dialogue multimodal emotion dataset, which is a valuable contribution to research on cross-cultural emotion analysis and recognition. Additionally, we conduct experiments on EmotionTalk to demonstrate the effectiveness and quality of the dataset. The EmotionTalk dataset will be made freely available for all academic purposes.

## 1 Introduction

Multimodal emotion recognition (MER) has become a key focus in artificial intelligence, integrating speech, vision, and text to capture the complexity of human emotions. It drives advancements in applications like virtual assistants, online education, and mental health monitoring. However, most research relies on English datasets, with Chinese resources remaining scarce. Existing datasets often face issues such as low quality, limited scale, and incomplete modalities, hindering model performance. Therefore, the development of a high-quality Chinese multimodal emotion recognition dataset is of critical importance to advance research in this field.

Traditional emotion recognition tasks include unimodal / multimodal emotion recognition on isolated utterances Liu et al. (2022; 2023); Sun et al. (2024a;b) and conversational emotion recognition Shi et al. (2020; 2023). The former relies on a single modality or integrates multimodal information for emotion recognition. For example, MISA Hazarika et al. (2020)utilizes modality-invariant and modality-specific representations to fuse multimodal information. DialogueRNN Majumder et al. (2019) extracts emotional information from conversations by modeling the speaker, context, and emotions within the dialogue. With the in-depth development of emotion recognition research, researchers have gradually introduced emerging tasks such as emotion captioning Xu et al. (2024); Liang et al. (2024b), driven by evolving application scenarios and practical demands. This task is first proposed by SECap Xu et al. (2024). Subsequently, this task has attracted increasing attention from researchers due to its unique value in interpretability, and has gradually become an important research direction in affective computing.

However, these studies use different datasets, and while they perform well in their respective experiments, directly comparing their performance remains challenging. This is mainly due to significant

differences in dataset scale, annotation methods, modality combinations, and dialogue structures, which affect model applicability and generalization. For instance, popular multimodal benchmarks like IEMOCAP Busso et al. (2008), MELD Poria et al. (2019), CMU-MOSEI Zadeh et al. (2018b), and CH-SIMS Yu et al. (2020) have been widely used but are primarily in English, with varying emotion category definitions and annotation standards, limiting cross-lingual and cross-cultural applicability. The underlying cause of this situation lies in the dilemma of data acquisition: on one hand, existing Chinese emotional data predominantly originates from film and television resources, which are relatively accessible but of low quality; on the other hand, compared to audiovisual materials, artificially recorded dialogue data can guarantee higher quality standards, thereby enabling the construction of datasets with greater academic research value. However, such high-quality controlled recording data is precisely what constitutes an extremely scarce resource at present. More seriously, emotion captioning research mostly relies on unpublished datasets, leading to a lack of standardized open benchmarks and further hindering research reproducibility and widespread application. Against the backdrop of current data scarcity, we deeply recognize that the importance of high-quality data has become increasingly prominent and cannot be overlooked.

To address these gaps, in this paper, we construct an large-scale interactive Chinese multimodal emotion dataset with fine-grained labels and emotional speaking style captions, **EmotionTalk**, in which the data are contributed by 19 professional actors, ensuring the naturalness and authenticity of the emotion expression. The dataset is in the form of dialogues, containing 23.6 hours of data and 19,250 utterances, along with corresponding labels that support various emotion tasks, including 7 discrete labels, 5 dimensional labels, and 4 caption labels. To the best of our knowledge, EmotionTalk is the first large-scale, comprehensive, recorded interactive Chinese multimodal emotion dataset. We further conduct experiments on unimodal emotion recognition, multimodal emotion recognition, and emotion caption tasks to validate the effectiveness and applicability of the constructed dataset. These experiments not only demonstrate the dataset's performance across different emotion tasks but also highlight its potential to support diverse model development and evaluation.

## 2 RELATED WORK

### 2.1 RELATED DATASETS

Table 1 presents the datasets which are commonly used in the field of multimodal emotion recognition, all of which consist of video, audio and text modalities.

**English Datasets:** The CMU-MOSEI Zadeh et al. (2018b) and MELD Poria et al. (2019) datasets provide large-scale multimodal data sourced from YouTube and TV shows, covering tasks such as discrete emotion classification and continuous sentiment intensity prediction. These datasets are advantageous due to their rich emotional labeling, but they are primarily derived from entertainment content, where emotional expressions tend to be exaggerated. As such, they may not fully capture the natural emotional expressions encountered in real-life situations. In contrast, the CREMA-D Cao et al. (2014), RAVDESS Livingstone & Russo (2018), IEMOCAP Busso et al. (2008) and MSP-IMPROV Busso et al. (2016) datasets are based on actor performances and emotion training, with IEMOCAP and MSP-IMPROV consist of conversational data, whereas CREMA-D and RAVDESS record non-dialogue data. These datasets offer higher-quality emotional data. However, these datasets largely rely on pre-written scripts, and their inherent limitations may lead to overly theatrical emotional expressions from actors, lacking the spontaneity found in authentic interactions.

**Chinese Datasets:** Currently, there have been some preliminary research efforts in the field of multimodal emotion datasets based on Mandarin For example, the CH-SIMS Yu et al. (2020) and MER-MULTI Lian et al. (2024) dataset use five continuous emotion labels and six discrete emotion labels respectively, making it suitable for multimodal sentiment analysis on isolated utterances spoken in Mandarin. However, both of them lack dialogue scenarios, overlooking the emotional changes multi-turn interactions. In contrast, datasets like M$^3$ED Zhao et al. (2022) and MC-EIU$_{ch}$ Liu et al. (2024) have made progress in terms of dialogue-level data, making it possible for supporting multimodal emotion recognition in conversations. Moreover, M$^3$ED and MC-EIU$_{ch}$ have been significant progress regarding the scale of the data.

Despite numerous advances in emotion recognition, most Chinese datasets still exhibit limitations in terms of scale, data quality, and annotation completeness. Existing Chinese datasets generally focus

Table 1: Summary of multimodal emotion datasets.

| Dataset | Modality | Dialogue | Sources | Emo-label | Des. | Language | Utts |
|---|---|---|---|---|---|---|---|
| CMU-MOSI Zadeh et al. (2016) | $a, v, l$ | No | YouTube | 7 Dim. | No | English | 2,199 |
| CMU-MOSEI Zadeh et al. (2018b) | $a, v, l$ | No | YouTube | 7 Disc. / 5 Dim. | No | English | 22,856 |
| MELD Poria et al. (2019) | $a, v, l$ | Yes | TVs | 7 Disc. | No | English | 13,708 |
| CREMA-D Cao et al. (2014) | $a, v, l$ | No | Act | 6 Disc. | No | English | 7,442 |
| RAVDESS Livingstone & Russo (2018) | $a, v, l$ | No | Act | 8 Disc. | No | English | 7,356 |
| IEMOCAP Busso et al. (2008) | $a, v, l$ | Yes | Act | 5 Disc. | No | English | 7,433 |
| MSP-IMPROV Busso et al. (2016) | $a, v, l$ | Yes | Act | 5 Disc. | No | English | 8,438 |
| CH-SIMS Yu et al. (2020) | $a, v, l$ | No | Movies, TVs | 5 Dim. | No | Mandarin | 2,281 |
| MER-MULTI Lian et al. (2024) | $a, v, l$ | No | Movies, TVs | 6 Disc. | No | Mandarin | 3,784 |
| M$^3$ED Zhao et al. (2022) | $a, v, l$ | Yes | TVs | 7 Disc. | No | Mandarin | 24,449 |
| MC-EIU_ch Liu et al. (2024) | $a, v, l$ | Yes | TVs | 7 Disc. | No | Mandarin | 11,003 |
| **EmotionTalk** | $a, v, l$ | **Yes** | **Record** | **7 Disc. / 5 Dim.** | **Yes** | **Mandarin** | **19,250** |

on relatively simple emotion labels or rely on low-quality data collected from the internet. In contrast, our dataset aims to effectively address these shortcomings by providing 23.6 hours of topic-driven spontaneous emotional dialogues. The dataset not only features high-quality recorded conversations but also includes detailed and comprehensive emotional annotations, making it a valuable asset for MER research and broader emotional dialogue analysis.

## 2.2 RELATED METHODS

### 2.2.1 MULTIMODAL EMOTION RECOGNITION

Multimodal emotion recognition involves identifying emotions from utterances or dialogues, where feature fusion methods are crucial. Recent advances include context representation modules for integrated multimodal features (Yang et al., 2023b), attention aggregation networks for cross-modal alignment (Fan et al., 2024), transformer-based models with self-distillation (Ma et al., 2024), and graph-based dynamic fusion networks (Hu et al., 2022). Additionally, continuous emotion recognition has gained attention, with frameworks using contrastive learning guided by sentiment intensity (Yang et al., 2024).

### 2.2.2 EMOTION CAPTIONING

To capture richer emotional information beyond traditional recognition, emotion captioning has emerged as a promising direction. Xu et al. (2024) propose a speech emotion captioning framework using LLaMA (Touvron et al., 2023) and HuBERT (Hsu et al., 2021a), while Liang et al. (2024a) design AlignCap to align captioning with human preferences, improving zero-shot generalization. Furthermore, Kawamura et al. (2024) demonstrate applications in TTS systems using speaker and speaking style captions, achieving enhanced naturalness and accuracy.

## 3 DATASET DESCRIPTION

In this section, we introduce a large-scale, comprehensive, recorded interactive Chinese multimodal emotion dataset, EmotionTalk. We describe data Collection, annotation and statistics in detail.

### 3.1 DATA COLLECTION

Compared to existing Chinese multimodal datasets, our unique approach lies in the data collection methodology. We moved away from traditional scripted methods, instead adopting a theme-driven improvisational performance approach with professional drama actors. This method aims to capture more authentic and natural emotional expressions, effectively simulating spontaneous emotional behavior in real-world scenarios. While this process is more challenging to implement and more time-consuming, its advantages in emotional authenticity are significant.

To ensure a high degree of data diversity, we developed multi-turn dialogue scenarios to simulate genuine human interaction. These scripts are inspired by television plots or generated by Large

Language Models (LLMs) and underwent rigorous manual quality assurance. Each scenario involves two characters, designed to capture the dynamic evolution of emotions over time. The scripts encompass a wide range of emotional intensities, from lighthearted conversations to tense, intense conflicts, fully showcasing the richness of emotional expression.

The dataset covers multiple real-life themes, including friendship, family, workplace, and doctor-patient interactions. The friendship theme includes dialogues about joy, conflict, and reconciliation, highlighting support and friction. The workplace theme involves complex emotions like collaboration, competition, pressure, and misunderstandings. Each theme is carefully designed; for instance, the family theme includes scenarios like arguments, holiday gatherings, and farewells, reflecting emotions like warmth, anger, and sadness. The language style varies by theme: friendship dialogues are casual and natural, while workplace exchanges are more formal and serious. This accurately simulates real-world language environments, encouraging actors to deliver authentic emotional performances and deepening the emotional layers and immersion of the scripts. It's important to note that actors are encouraged to express genuine emotions based on the theme rather than adhering strictly to a script. To ensure emotional consistency and avoid actors maintaining a single emotion for too long, each dialogue is limited to approximately two minutes.

## 3.2 ANNOTATION

To ensure the high quality and diversity of the dataset, we design a rigorous data annotation process, incorporating multi-dimensional annotations for emotion categories, emotion intensity, and emotional speaking style caption. The detailed annotation process is outlined below:

**Emotion Category:** For each sample, we design a multi-step annotation process with cross-validation by $N$ ($N = 5$) annotators. The emotion category annotation is based on the basic emotion theory commonly used in psychological research and covers $K$ ($K = 7$) widely recognized emotion categories: happiness, surprise, sadness, disgust, anger, fear, and neutral. To prevent interference between different modalities and avoid potential confusion, we follow the modality-independent annotation principle, requiring annotators to view only the current modality information and strictly prohibiting multimodal synchronous annotation. The annotation process follows a predefined sequence, processing text, audio, silent video, and finally video with audio in order. Each emotion annotation consists of a emotion category $y_i$ and a confidence score $c_i$, of which is set to 0.1, 0.3, 0.5, 0.7, and 0.9, to quantify the annotator's confidence in their judgment. The formula for calculating the weighted confidence score $x_k$, $k = \{1, \ldots, K\}$ for each category of a sample is as follows:

$$x_k = \frac{1}{N_k} \sum_{i=1}^{N} \mathbb{I}(y_i = k) \cdot c_i, \tag{1}$$

where $k$ represents the emotion category, $N_k$ is the number of annotations for category $k$, $\mathbb{I}(y_i = k)$ is an indicator function, which equals 1 if the label $y_i$ assigned by annotator $i$ is equal to category $k$, and 0 otherwise.

Thus, the final emotion category $y$ is calculated as follows:

$$y = \arg \max_k x_k, \tag{2}$$

where $argmax$ represents selecting the category $k$ that corresponds to the maximum weighted confidence $x_k$ as the final category label.

For cases with low confidence or inconsistent annotations, we employ a multi-round negotiation mechanism: first, multiple experienced annotators independently re-evaluate the samples, followed by expert discussions to reach consensus, ensuring the reliability and consistency of annotation quality.

**Emotion Intensity:** To more accurately quantify the intensity of emotional expressions, we have designed a multimodal-based emotion intensity annotation process aimed at quantitatively labeling the emotional polarity (positive, negative, neutral) and its intensity in utterances. For each audio clip, five annotators will be assigned, and each annotator will evaluate the emotional state as -2 (strongly negative), -1 (weakly negative), 0 (neutral), 1 (weakly positive), or 2 (strongly positive). The annotation results from the five annotators are then averaged to obtain a continuous label that contains emotion intensity information. The final labeling results will be one of the following values: {-2.0, -1.8, -1.6, -1.4, -1.2, -1.0, -0.8, -0.6, -0.4, -0.2, 0.0, 0.2, 0.4, 0.6, 0.8, 1.0, 1.2, 1.4, 1.6, 1.8, 2.0}.

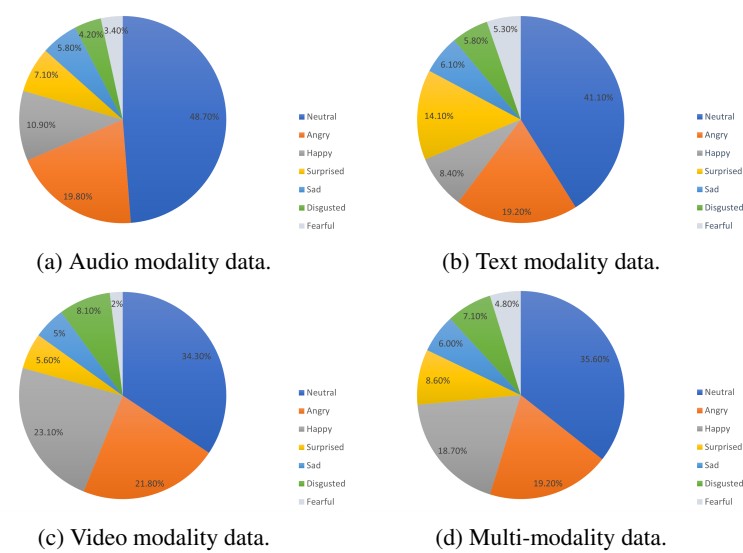

Figure 1: Data distribution across different modalities.

Smaller values indicate higher negativity, while larger values indicate higher positivity. Similarly, for samples with conflicting emotion polarity annotations, we adopt a multi-round negotiation mechanism: 2-3 senior emotion annotation experts engage in thorough discussions to negotiate and reach final decisions on disputed samples, ensuring the authority and consistency of annotation results.

**Emotional Speaking Style Caption:** A core innovation of this dataset lies in constructing a four-dimensional, fine-grained speech emotion annotation system that achieves significant breakthroughs in annotation comprehensiveness, precision, and innovation. Our proposed annotation framework encompasses four refined dimensions: speaker, speaking style, emotion, and overall comprehensive description. In terms of comprehensiveness, the system constructs a complete spectrum of speech feature descriptions, ranging from basic vocal qualities (such as warmth, richness, and clarity) to high-level speaking styles (such as speech rhythm, intonation patterns, and pause structures), and further to deep emotional semantics (such as emotion types and intensity levels). Regarding fine-grained granularity, we conduct in-depth deconstruction of each dimension, decomposing complex speech phenomena into quantifiable and describable microscopic feature units. In terms of innovation, we propose for the first time a comprehensive modeling approach for multi-dimensional features, semantically fusing the three dimensions of vocal qualities, speaking styles, and emotions to generate unified holistic descriptive annotations. Finally, we employ the deepSeek-R1 Guo et al. (2025) large language model to intelligently expand the overall descriptions, generating five annotation variants that maintain semantic consistency while exhibiting diverse expressive styles. This not only significantly enriches the expressive diversity of the dataset but also provides unprecedented fine-grained annotation resources for multimodal emotion understanding.

## 3.3 STATISTICS

Our dataset comprises 744 dialogues with a total of 19,250 unimodal samples covering text, audio, and video modalities. The audio data spans 23.6 hours with an average segment length of 4.4 seconds, while the text data contains 469,387 characters with approximately 24 characters per sentence on average. To support comprehensive emotion analysis, we provide independent and detailed emotion labels for each modality (text, audio, video) while constructing multimodal fusion labels, forming a hierarchically rich and comprehensively covered annotation framework.

In our dataset, each modality exhibits distinct emotional distribution characteristics, which validates the importance of multimodal perception. In the audio modality, neutral emotions (48.7%) and anger emotions (19.8%) constitute the primary components, while various emotion categories including happiness, surprise, sadness, disgust, and fear are also sufficiently represented. In the text modality, neutral, anger, and surprise are the dominant emotion categories, accounting for 41.1%, 19.2%, and

14.1% respectively. The video modality displays an even richer emotional distribution. In video-only scenarios, neutral, happiness, and anger are the three most prevalent categories, accounting for 34.3%, 23.1%, and 21.8% respectively. When multimodal, emotional perception undergoes significant changes: neutral, anger, and happiness account for 35.6%, 19.2%, and 18.7% respectively. This variation clearly demonstrates the unique impact of different modalities on emotional perception, and the distribution effectively reflects emotional expression patterns in real conversations, providing rich sample diversity for model training.

It is particularly noteworthy that the same sample may exhibit different emotion labels across different modalities, a phenomenon that fully demonstrates the complexity and necessity of multimodal emotion understanding. For instance, the textual content of an utterance may convey neutral emotion, while its vocal intonation transmits anger, and facial expressions may display sadness. This cross-modal emotional inconsistency precisely reflects the authenticity and multi-layered nature of human emotional expression, proving the limitations of relying solely on a single modality for emotion recognition and highlighting the important value of multimodal fusion analysis. The cross-modal emotion label variance analysis indicates that our dataset not only provides rich emotion category coverage but, more importantly, reveals the unique roles and complementary values of different modalities in emotional communication. This design enables researchers to deeply explore inter-modal emotional consistency and conflicts, laying a solid foundation for developing more robust and accurate multimodal emotion recognition models.

### 3.4 Inter-rater Agreement Analysis

Fleiss' Kappa is a statistical measure assessing inter-rater agreement when multiple evaluators categorize items. The Kappa value ranges from -1 to 1, where values of 0.41-0.60 indicate moderate agreement, 0.61-0.80 reflect substantial agreement, and 0.81-1.00 indicate almost perfect agreement. The formula for Fleiss' Kappa is as follows:

$$\kappa = \frac{P_o - P_e}{1 - P_e},\tag{3}$$

where $P_o$ represents the observed proportion of agreement, and $P_e$ represents the expected proportion of agreement under random conditions.

In our dataset, the Fleiss' Kappa values are 0.79 for audio, 0.66 for text, 0.73 for video without audio, and 0.78 for video with audio, indicating substantial to almost perfect agreement across all modalities.

## 4 Experiments

In this section, we evaluate our dataset across a variety of tasks, including unimodal emotion recognition, multimodal emotion recognition, multimodal emotion analysis, and emotional speaking style captioning. We build our experimental pipeline upon the MerBench Lian et al. (2024), which provides a standardized setup for benchmarking multimodal models.

Specifically, in the continuous setting, we focus on a binary classification task that distinguishes between positive and negative emotions, where samples with scores below 0 are labeled as negative, and those above 0 as positive. For the first three tasks, accuracy (ACC) is used as the primary evaluation metric, while for the speaker emotion-style captioning task, we adopt $BLEU_4$, $ROUGE_L$, METEOR, SPIDEr, FENSE, BERTScore and CLAPScore for evaluation. To facilitate reproducibility, we document all experimental settings in Appendix B.2, including hyperparameter tuning strategies, optimizer selection, and the values of all key training parameters.

### 4.1 Unimodal Emotion Recognition

This section reports the emotion recognition performance of different feature extractors on the corresponding modalities, as shown in Table 2.

**Feature Extractor:** To assess the performance of our dataset, we employ a comprehensive suite of pre-trained baseline models across different modalities. Specifically, for the speech modality, we utilize Wav2Vec 2.0 Baevski et al. (2020), HuBERT Hsu et al. (2021b), WavLM Chen et al. (2022),

Table 2: We report unimodal results for the EmotionTalk dataset. Four means that four emotion labels are used: happy, angry, sad, and neutral. All means that all emotion labels are used.

| Speech Model | Speech(Four) | Multimodal(Four) | Speech(All) | Multimodal(All) | Mean |
|---|---|---|---|---|---|
| Whisper-Base | 71.03 | 60.44 | 56.61 | 48.47 | 59.14 |
| Whisper-Large | 75.45 | 61.90 | 60.34 | 49.56 | 61.81 |
| WavLM-Base | 72.50 | 62.96 | 59.72 | 53.14 | 62.08 |
| Wav2vec 2.0-Base | 77.31 | 63.85 | 62.16 | 50.96 | 63.57 |
| Wav2vec 2.0-Large | 76.22 | 64.68 | 63.14 | 51.06 | 63.78 |
| WavLM-Large | 76.67 | 64.48 | 61.90 | 53.91 | 64.24 |
| Hubert-Large | **82.88** | **73.69** | 66.15 | 61.12 | 70.96 |
| Hubert-Base | 81.09 | 73.09 | **68.64** | **62.52** | **71.34** |
| **Text Model** | **Text(Four)** | **Multimodal(Four)** | **Text(All)** | **Multimodal(All)** | **Mean** |
| Vicuna-7B | 55.24 | 46.26 | 45.57 | 43.91 | 47.75 |
| LERT-Base | 59.68 | 51.36 | 46.09 | 38.26 | 48.85 |
| DeBERTa-Large | 57.46 | 49.11 | 44.89 | 44.79 | 49.06 |
| BERT-Base | 57.66 | 50.83 | 46.50 | 44.69 | 49.92 |
| Sentence-BERT | 56.52 | 52.15 | 46.45 | 45.05 | 50.04 |
| BLOOM-7B | 60.87 | 50.56 | 47.38 | 43.23 | 50.51 |
| ChatGLM2-6B | **60.95** | 55.47 | 46.19 | 41.16 | 50.94 |
| RoBERTa-Large | 59.48 | 53.88 | 46.86 | 44.27 | 51.12 |
| RoBERTa-Base | 60.15 | 50.96 | 48.11 | **45.52** | 51.19 |
| Baichuan-7B | 60.08 | **56.39** | **48.21** | 41.84 | **51.63** |
| **Visual Model** | **Visual(Four)** | **Multimodal(Four)** | **Visual(All)** | **Multimodal(All)** | **Mean** |
| Data2vec-Base | 35.72 | 29.69 | 40.44 | 32.92 | 34.69 |
| VideoMAE-Base | 54.18 | 47.51 | 54.33 | 46.29 | 50.58 |
| EVA-02-Base | 69.87 | 54.27 | 58.84 | 38.88 | 55.47 |
| VideoMAE-Large | 62.36 | 64.74 | 55.68 | 50.54 | 58.33 |
| CLIP-Base | 71.38 | 63.95 | 59.51 | 49.09 | 60.98 |
| Dinov2-Large | 70.60 | 68.99 | 60.96 | **54.59** | 63.79 |
| Dinov2-Giant | 73.42 | 69.58 | 62.73 | 53.76 | 64.87 |
| CLIP-Large | **77.81** | **73.96** | **64.75** | 54.17 | **67.67** |

and Whisper Radford et al. (2023). For the text modality, our selection includes Vicuna-7B Chiang et al. (2023), LERT Cui et al. (2022), DeBERTa He et al. (2020), BERT Devlin et al. (2019), Sentence-BERT Reimers & Gurevych (2019), BLOOM-7B Workshop et al. (2022), RoBERTa Liu et al. (2019), ChatGLM2 Du et al. (2021) and Baichuan-7B Yang et al. (2023a). For the visual modality, we adopt Data2Vec Baevski et al. (2022), VideoMAE Tong et al. (2022), EVA-02 Fang et al. (2024), CLIP Radford et al. (2021), and DINOv2 Oquab et al. (2023).

Based on the comparative performance of these encoders shown in Table 2, we aim to provide guidance for modality-specific feature selection in downstream emotion recognition tasks. Given that the EmotionTalk dataset provides independent unimodal annotations, we conduct two experimental settings to investigate the capability of unimodal representations in emotion recognition. In the first setting, we utilize ground-truth unimodal labels to evaluate each model's ability to perform unimodal emotion classification. In the second setting, we adopt multimodal labels instead, to assess whether a single modality alone can reliably infer the speaker's actual emotional state.

Several key findings emerge from these experiments. First, for the same unimodal classification task, models consistently achieve better performance when trained and evaluated with unimodal labels than with multimodal labels. This suggests that models are effective at capturing the modality-specific emotional cues. However, these results do not necessarily reflect the speaker's actual emotional state, as unimodal annotations may be biased or incomplete. This indicates that unimodal information remains a valuable signal for emotion recognition, though it is inherently limited in expressiveness and scope. Consequently, relying solely on unimodal representations is insufficient for accurately

Table 3: We report multimodal results for the EmotionTalk dataset. Four means that only four emotion labels are used: happy, angry, sad, and neutral. All means that all of the emotion labels are used.

| Features | Algorithms | Fusion | Multimodal(Four) | Multimodal(All) | Mean |
|---|---|---|---|---|---|
| | MCTN | Frame-level | 65.34 | 47.80 | 56.57 |
| | MFM | Frame-level | 75.94 | 59.51 | 67.73 |
| Hubert-Base | GMFN | Frame-level | 76.87 | 63.66 | 70.27 |
| | MMIM | Uttrance-level | 78.93 | 64.54 | 71.74 |
| Baichuan-7B | MISA | Uttrance-level | 80.58 | 66.77 | 73.68 |
| | TFN | Uttrance-level | 80.12 | 68.27 | 74.20 |
| CLIP-Large | MulT | Frame-level | 82.17 | 66.67 | 74.42 |
| | MFN | Frame-level | 80.38 | **69.31** | 74.85 |
| | Attention | Uttrance-level | 82.11 | 68.17 | 75.14 |
| | LMF | Uttrance-level | **81.31** | 69.10 | **75.21** |

Table 4: "Top4" indicates that we select the top 4 models for each modality (their ranking is based on the results in Table 2). We utilize the LMF for multimodal fusion.

| | | | Multimodal | | | |
|---|---|---|---|---|---|---|
| # Top | Text | Speech | Visual | Discrete(Four) | Discrete(All) | Continuous | Mean |
| Top 1 | Baichuan-7B | Hubert-Base | CLIP-Large | 81.31 | 69.10 | **93.35** | 81.25 |
| Top 2 | RoBERTa-Base | Hubert-Large | Dinov2-Giant | **83.23** | **69.21** | 93.16 | **81.87** |
| Top 3 | RoBERTa-Large | WavLM-Large | Dinov2-Large | 78.13 | 65.01 | 93.10 | 78.75 |
| Top 4 | ChatGLM2-6B | W2v 2.0-Large | CLIP-Base | 73.82 | 63.50 | 92.26 | 76.53 |

capturing complex emotional states, reinforcing the importance of multimodal fusion in emotion understanding.

## 4.2 MULTIMODAL EMOTION RECOGNITION / SENTIMENT ANALYSIS

Table 3 presents the performance of various multimodal fusion algorithms on the EmotionTalk dataset using the optimal encoder from each modality—HuBERT-Base (speech), Baichuan-7B (text), and CLIP-Large (visual). The fusion methods are categorized into frame-level (e.g., MFN Zadeh et al. (2018a), GMFN Zadeh et al. (2018b), MCTN Pham et al. (2019), MFM Tsai et al. (2018), and MulT Tsai et al. (2019)) and utterance-level (e.g., TFN Zadeh et al. (2017), LMF Liu et al. (2018), MISA Hazarika et al. (2020), MMIM Han et al. (2021), and the Attention mechanism Vaswani et al. (2017)) strategies, enabling a comparative analysis of their effectiveness in multimodal emotion recognition.

Several important observations can be drawn. First, utterance-level fusion methods generally outperform frame-level approaches in both the four-class and full-class emotion classification settings. For instance, LMF achieves the highest score in the Multimodal(Four) setting (83.04%) and also yields the best average performance (75.53%), indicating that aligning features at the utterance level better captures the holistic emotional state. Similarly, attention-based fusion also performs competitively, with an average score of 75.14%, suggesting the advantage of adaptive weighting across modalities. Moreover, due to the limited scale of emotion datasets, complex fusion algorithms are prone to overfitting. In contrast, simple yet effective fusion strategies often achieve relatively better performance.

Table 4 reports the multimodal emotion recognition results using the top four models from each modality, selected based on unimodal performance in Table 2. All combinations adopt the LMF algorithm for fusion. Among the configurations, the combination of RoBERTa-Base (text), HuBERT-Large (speech), and Dinov2-Giant (visual) achieves the best overall performance, with the highest score in the Discrete (Four) setting (83.23%) and the highest average (81.87%). Notably, different model combinations yield comparable performance on the continuous labels, while their results on

Table 5: Automatic captioning results. All methods use Hubert as the speech encoder.

| | Decoder | BLEU$_4$ | ROUGE$_L$ | METEOR | SPIDEr | FENSE | BERTScore | CLAPScore |
|---|---|---|---|---|---|---|---|---|
| Speaker | Transformer-based | 0.011 | 0.397 | 0.204 | 0.229 | 0.842 | 0.974 | 0.860 |
| | GPT-2 | **0.020** | **0.430** | **0.212** | 0.256 | 0.765 | 0.976 | **0.899** |
| | Qwen-2 | 0.009 | 0.414 | 0.205 | **0.258** | **0.846** | **0.977** | 0.878 |
| Style | Transformer-based | 0.065 | 0.517 | 0.313 | 0.339 | 0.512 | 0.985 | 0.895 |
| | GPT-2 | 0.075 | 0.510 | 0.298 | 0.350 | **0.611** | 0.987 | 0.850 |
| | Qwen-2 | **0.127** | **0.564** | **0.339** | **0.482** | 0.523 | **0.988** | **0.912** |
| Emotion | Transformer-based | 0.032 | 0.366 | 0.191 | 0.276 | 0.932 | 0.973 | 0.843 |
| | GPT-2 | 0.014 | **0.399** | 0.147 | 0.235 | 0.903 | 0.972 | 0.818 |
| | Qwen-2 | **0.058** | 0.361 | **0.199** | **0.353** | **0.942** | **0.975** | **0.853** |
| Overall | Transformer-based | 0.018 | 0.469 | 0.233 | **0.230** | **0.921** | 0.980 | 0.878 |
| | GPT-2 | 0.015 | 0.462 | 0.214 | 0.227 | 0.890 | 0.980 | 0.849 |
| | Qwen-2 | **0.033** | **0.535** | **0.268** | 0.121 | 0.562 | **0.984** | **0.885** |

discrete tasks vary considerably, underscoring the impact of feature selection. These findings confirm that even under the same fusion strategy, the choice of multimodal features can significantly affect the overall performance of multimodal fusion.

### 4.3 EMOTIONAL SPEAKER STYLE CAPTIONING

Table 5 presents a comprehensive comparison of three decoder architectures—Transformer-based, GPT-2, and Qwen-2—across multiple captioning dimensions (Speaker, Style, Emotion, and Overall), evaluated using a suite of standard automatic metrics. Qwen-2 outperforms other models across all four tasks, demonstrating its effectiveness in producing captions that preserve both emotional nuance and stylistic diversity. The strong BERTScore suggests that its generation aligns closely with human references at the semantic level, beyond surface-level lexical similarity. Although Qwen-2 achieved the best overall performance, GPT-2 performed notably well on the speaker-focused task, obtaining the highest ROUGE$_L$ (0.430) and CLAPScore (0.899). In contrast, the Transformer-based decoder showed weaker overall results but maintained basic structural coherence and content coverage, as reflected in its ROUGE$_L$ and SPIDEr scores. Overall, these results highlight Qwen-2's robustness in capturing fine-grained stylistic and emotional cues, crucial for emotional speaking style captioning.

## 5 CONCLUSION

This paper presents EmotionTalk—a high-quality Chinese conversational multimodal emotion dataset with rich annotations. Unlike existing datasets, EmotionTalk provides 23.6 hours of multimodal conversational data recorded by 19 professional actors through topic-guided spontaneous emotional dialogues, ensuring the authenticity and naturalness of emotional expressions. EmotionTalk not only fills the dual gap of high-quality recorded data and emotional speaking style description annotation in Chinese multimodal emotion research, but also becomes a valuable resource in the field of affective computing through its advantages of interactive recording, multimodality, and rich annotations. The quality of the dataset is validated through rigorous data creation processes and extensive diverse baseline experiments. EmotionTalk provides a benchmark testing platform for future multimodal emotion recognition and emotional dialogue modeling, aiming to advance the development of affective computing and human-computer interaction fields.

## 6 ETHICS STATEMENT

This study is conducted in accordance with rigorous ethical guidelines to ensure the protection of participants' rights and well-being. All recordings take place in a quiet indoor environment, where professional actors engage in natural, emotionally diverse, and logically coherent dialogues based on

predefined emotional themes and content outlines. The annotation cost for each data sample is 0.2 RMB.

To preserve participant privacy, all data are anonymized by removing personal identifiers and replacing them with coded labels. The dataset is released under the CC BY-NC 4.0 license, which prohibits commercial use and supports ethical research practices. Data are securely stored, and access is restricted to authorized researchers for academic purposes only.

In conclusion, this study demonstrates a strong commitment to ethical standards, encompassing informed consent, the protection of personal privacy, appropriate compensation, and the responsible dissemination of data, thereby safeguarding participant rights and supporting ethical scientific advancement.

## 7 REPRODUCIBILITY STATEMENT

All experiments in this study strictly adhere to the principles of reproducibility. To facilitate replication by future researchers, we have provided a detailed list in the appendix B.1 and B.2, of all hyperparameter settings, as well as the precise source (including version numbers and reference links) for each model and feature extraction tool. All code has been open-sourced to ensure the transparency and verifiability of our results.

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

# A DATASHEETS FOR DATASETS

## A.1 DATASET SNAPSHOTS

The dataset comprises 744 dialogues, encompassing a total of 19,250 utterances for each unimodal modality—text, audio, and video. The audio data span approximately 23.6 hours, with an average duration of 4.4 seconds per utterance.

Each utterance is stored as an individual JSON file following a unique naming convention in the format: `<group_No>_<session_No>_<Speaker_id>_<Utt_No>.json`. Corresponding audio and video files are named identically, with the extensions ".wav" and ".mp4" respectively: `<group_No>_<session_No>_<Speaker_id>_<Utt_No>.wav` and `<group_No>_<session_No>_<Speaker_id>_<Utt_No>.mp4`. The samples of audio and video files in EmotionTalk are shown in Fig. 2.

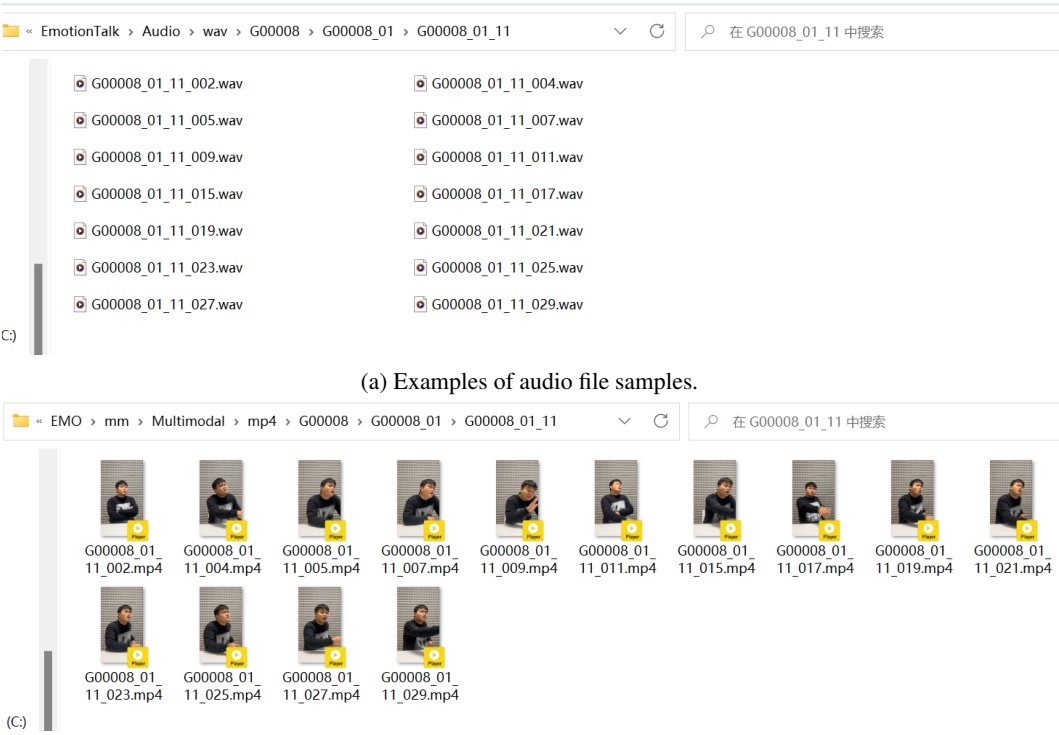

(a) Examples of audio file samples.

(b) Examples of video file samples.

Figure 2: Snapshots of audio and video samples in the EmotionTalk dataset. All files are named following a consistent and structured format.

## A.2 DATA FORMAT

Each utterance in the EmotionTalk dataset is associated with a corresponding ".jsonl" file, which contains detailed sample-level annotations. These annotations include not only the basic information such as the emotion label, speaker identity, and transcript, but also rich metadata that describes the expressive characteristics of the utterance. The detailed annotation fields are listed in Table 6.

## A.3 DATA DISTRIBUTION

In this study, to make full use of the data and ensure both effective model training and fair evaluation, the dataset is divided into training, validation, and test sets in a approximate ratio of 8:1:1. Specifically, 80% of the data is used for training the model to learn effective feature representations, 10% is allocated for validation to assist in model selection and prevent overfitting during training, and the

remaining 10% is reserved as the test set to evaluate the model's generalization performance. When splitting the dataset, we make effort to ensure that the data distribution of each category remains consistent across the training, validation, and test sets. A detailed information of the the distribution across different subsets is presented in the Table 7.

Table 6: Description of Sample-Level Annotations

| Name | Description |
|---|---|
| emotion | Emotion label.[1] |
| Confidence_degree | Annotator's self-rated confidence in the emotion label. |
| Continuous_label | 5-dimensional sentiment labels.[2] |
| speaker_id | Unique speaker identifier. |
| emotion_result | Final aggregated emotion label.[3] |
| Continuous label_result | Final averaged sentiment labels aggregated from five annotators. |
| content | Transcript of the utterance. |
| startTime | Utterance start time in the session. |
| endTime | Utterance end time in the session. |
| duration | Total duration of the utterance. |
| emo_cap | Caption describing the type and intensity of the expressed emotion. |
| spe_cap | Caption describing the speaker's voice quality. |
| style_cap | Caption describing speaking style. |
| caption_1 – caption_5 | Emotional speaking style caption. |
| file_path | Relative path to the audio file. |

[1] The emotion categories include: happiness, surprise, sadness, disgust, anger, fear, and neutral.
[2] The 5-dimensional sentiment labels include: -2 (strongly negative), -1 (weakly negative), 0 (neutral), 1 (weakly positive), or 2 (strongly positive).
[3] The computation method is detailed in Section 3.2 Annotation.

Table 7: Statistics of the data distribution across the training, validation, and test sets.

| | Angry | Disgusted | Fearful | Happy | Neutral | Sad | Surprised | Total |
|---|---|---|---|---|---|---|---|---|
| Train | 2950 | 1142 | 672 | 2986 | 5377 | 919 | 1367 | 15413 |
| Validation | 409 | 95 | 125 | 360 | 675 | 111 | 133 | 1908 |
| Test | 339 | 134 | 125 | 246 | 801 | 123 | 161 | 1929 |
| Total | 3698 | 1371 | 922 | 3592 | 6853 | 1153 | 1661 | 19250 |

## B  FEATURE EXTRACTION

### B.1  MODELS

To comprehensively evaluate the proposed dataset, we conduct extensive experiments on three tasks: unimodal emotion recognition, multimodal emotion recognition / sentiment analysis and emotional speaker style captioning. For the unimodal and multimodal emotion recognition tasks, we employ a range of state-of-the-art models as feature extractors to obtain representations from each modality. Then, we select several high-quality features as the foundation for multimodal fusion. For the emotional speaker style captioning task, we utilize three types of decoders, including transformer, GPT-2 and Qwen-2, to assess the quality and utility of the dataset. The details of the models are provided in Table 8.

Table 8: An overview of the models employed across different tasks.

| Speech Model | Link | License |
|---|---|---|
| Whisper-Base Radford et al. (2023) | huggingface.co/openai/whisper-base | Apache License 2.0 |
| Whisper-Large Radford et al. (2023) | huggingface.co/openai/whisper-large-v2 | Apache License 2.0 |
| WavLM-Base Chen et al. (2022) | huggingface.co/microsoft/wavlm-base | CC BY-SA 3.0 |
| Wav2vec 2.0-Base Baevski et al. (2020) | huggingface.co/TencentGameMate/chinese-wav2vec2-base | MIT License |
| Wav2vec 2.0-Large Baevski et al. (2020) | huggingface.co/TencentGameMate/chinese-wav2vec2-large | MIT License |
| WavLM-Large Chen et al. (2022) | huggingface.co/microsoft/wavlm-large | CC BY-SA 3.0 |
| Hubert-Large Hsu et al. (2021b) | huggingface.co/TencentGameMate/chinese-hubert-large | MIT License |
| Hubert-Base Hsu et al. (2021b) | huggingface.co/TencentGameMate/chinese-hubert-base | MIT License |

| Text Model | Link | License |
|---|---|---|
| Vicuna-7B Chiang et al. (2023) | huggingface.co/CarperAI/stable-vicuna-13b-delta | CC BY-NC-SA 4.0 |
| LERT-Base Cui et al. (2022) | huggingface.co/hfl/chinese-lert-base | Apache License 2.0 |
| DeBERTa-Large He et al. (2020) | huggingface.co/microsoft/deberta-v3-large | MIT License |
| BERT-Base Devlin et al. (2019) | huggingface.co/google-bert/bert-base-chinese | Apache License 2.0 |
| Sentence-BERT Reimers & Gurevych (2019) | huggingface.co/sentence-transformers/paraphrase-multilingual-mpnet-base-v2 | Apache License 2.0 |
| BLOOM-7B Workshop et al. (2022) | huggingface.co/bigscience/bloom-7b1 | BigScience Responsible AI License 1.0 |
| ChatGLM2-6B Du et al. (2021) | huggingface.co/THUDM/chatglm2-6b | Apache License 2.0 |
| RoBERTa-Large Liu et al. (2019) | huggingface.co/hfl/chinese-roberta-wwm-ext-large | Apache License 2.0 |
| RoBERTa-Base Liu et al. (2019) | huggingface.co/hfl/chinese-roberta-wwm-ext | Apache License 2.0 |
| Baichuan-7B Yang et al. (2023a) | huggingface.co/baichuan-inc/Baichuan-7B | |

| Visual Model | Link | License |
|---|---|---|
| Data2vec-Base Baevski et al. (2022) | huggingface.co/facebook/data2vec-vision-base | Apache License 2.0 |
| VideoMAE-Base Tong et al. (2022) | huggingface.co/MCG-NJU/videomae-base | CC BY-NC 4.0 |
| EVA-02-Base Fang et al. (2024) | https://huggingface.co/timm/eva02_base_patch14_224.mim_in22k | MIT License |
| VideoMAE-Large Tong et al. (2022) | huggingface.co/MCG-NJU/videomae-large | CC BY-NC 4.0 |
| CLIP-Base Radford et al. (2021) | huggingface.co/openai/clip-vit-base-patch32 | Apache License 2.0 |
| Dinov2-Large Oquab et al. (2023) | huggingface.co/facebook/dinov2-large | Apache License 2.0 |
| Dinov2-Giant Oquab et al. (2023) | huggingface.co/facebook/dinov2-giant | Apache License 2.0 |
| CLIP-Large Radford et al. (2021) | huggingface.co/openai/clip-vit-large-patch14 | Apache License 2.0 |

| Captioning Model | Link | License |
|---|---|---|
| Transformer-based Lewis et al. (2019) | huggingface.co/fnlp/bart-base-chinese | Apache License 2.0 |
| GPT-2 Lagler et al. (2013) | huggingface.co/uer/gpt2-chinese-cluecorpussmall | Apache License 2.0 |
| Qwen-2 Yang et al. | huggingface.co/Qwen/Qwen2-7B | Apache License 2.0 |

Table 9: Training hyperparameters used for the unimodal and multimodal models in Table 2 on the EmotionTalk dataset. "Four" refers to using four emotion labels (happy, angry, sad, neutral), while "All" refers to using the full label set.

| Hyperparameter | Four (Unimodal/Multimodal) | All (Unimodal/Multimodal) |
|---|---|---|
| Learning Rate | 1e-3 | 1e-5 |
| L2 Regularization Weight | 1e-5 | 1e-5 |
| Batch Size | 32 | 32 |
| Epochs | 100 | 100 |

## B.2 HYPERPARAMETERS AND COMPUTING RESOURCES

We provide open access to both the data and the code used in our experiments. The full experimental code is available at https://github.com/NKU-HLT/EmotionTalk. Key training hyperpa-

rameters for different models are summarized in Table 9, Table 10, and Table 11. All models are trained using the AdamW optimizer.

The experiments based on the Qwen-2 decoder are conducted on an NVIDIA A800 GPU, while all other experiments are performed using an NVIDIA GeForce RTX 3090 GPU.

Table 10: Key training hyperparameters used for each multimodal model in Table 3 on the EmotionTalk dataset.

| Model | Hidden Dim | Dropout | Learning Rate | Grad Clip |
|---|---|---|---|---|
| MCTN | 64 – 256 | 0.0 – 0.3 | 1e-3 | 0.6 – 1.0 |
| MFM | 128 / 256 | 0.0 – 0.7 | 1e-3 | -1.0 |
| GMFN | 128 / 256 | 0.0 – 0.7 | 1e-3 | -1.0 |
| MMIN | 64 – 256 | 0.0 – 0.3 | 1e-3 | 0.6 – 1.0 |
| MISA | 64 – 256 | 0.2 – 0.5 | 1e-4 | -1.0 – 1.0 |
| TFN | 64 / 128 | 0.2 – 0.5 | 1e-3 | -1.0 |
| MulT | 64 – 256 | 0.0 – 0.3 | 1e-3 | 0.6 – 1.0 |
| MFN | 128 / 256 | 0.0 – 0.7 | 1e-3 | -1.0 |
| Attention | 64 – 256 | 0.2 – 0.5 | 1e-5 | -1.0 |
| LMF | 32 – 256 | 0.2 – 0.5 | 1e-5 | -1.0 |

Table 11: Training hyperparameters for each decoder in Table 5.

| Decoder | Batch Size | Epochs | Learning Rate | Weight Decay | Warmup |
|---|---|---|---|---|---|
| Transformer-based | 8 | 15 | 1.7e-05 | 3.0e-04 | 0 |
| GPT-2 | 8 | 15 | 1.7e-05 | 3.0e-04 | 0 |
| Qwen-2 | 4 | 6 | 1e-4 | 0.0 | 1,000 |

## C  ANNOTATION WEBSITE

To improve the efficiency of the annotation process, we conduct data annotation and quality assessment on a data platform. As shown in the Fig 3, this platform supports the annotation of various tasks such as speech emotion recognition and emotional speaking style captioning.

## D  EXTRA EXPERIMENT RESULTS AND ANALYSIS

Table 12: Accuracy (%) comparison of different models across modalities (Speech, Text, and Speech+Text) on the Emotion Prediction in Conversation (EPC) task.

| Modality | Model | ACC |
|---|---|---|
| Speech | BiGRU | 62.58 |
| | AVEF | 60.81 |
| | DEP | 65.40 |
| | EAMT | **65.85** |
| Text | BiGRU | 44.19 |
| | AVEF | **49.92** |
| | DEP | 49.53 |
| | EAMT | 48.68 |
| Speech+Text | BiGRU | **65.70** |
| | AVEF | 64.47 |
| | DEP | 65.25 |
| | EAMT | 65.47 |

Table 13: Accuracy (%) comparison of different models across modalities (Speech and Text) on the Emotion Recognition in Conversation (ERC) task.

| Modality | Model | ACC |
|---|---|---|
| Speech | CMN | 66.37 |
| | ICON | 65.31 |
| | DialogueRNN | 66.34 |
| | DialoguGCN | **67.51** |
| Text | CMN | 46.67 |
| | ICON | 47.38 |
| | DialogueRNN | 49.63 |
| | DialoguGCN | **49.75** |

In the task of Emotion Prediction in Conversation (EPC), we observe clear performance differences across modalities and models Shahriar & Kim (2019); Shi et al. (2020; 2023), as shown in Table 12. Speech-based models consistently outperform text-based ones, highlighting the importance of vocal information in anticipating upcoming emotional states. Among the speech-only models, EAMT Shi et al. (2023) achieves the highest accuracy at 65.85%, marginally surpassing DEP Shi et al. (2020) (65.40%) and BiGRU (62.58%), indicating the benefit of explicitly modeling multi-role contextual information. Text-only models perform significantly worse, with BiGRU scoring only 44.19%, and EAMT, despite its contextual modeling, reaching just 48.68%. Interestingly, multimodal fusion

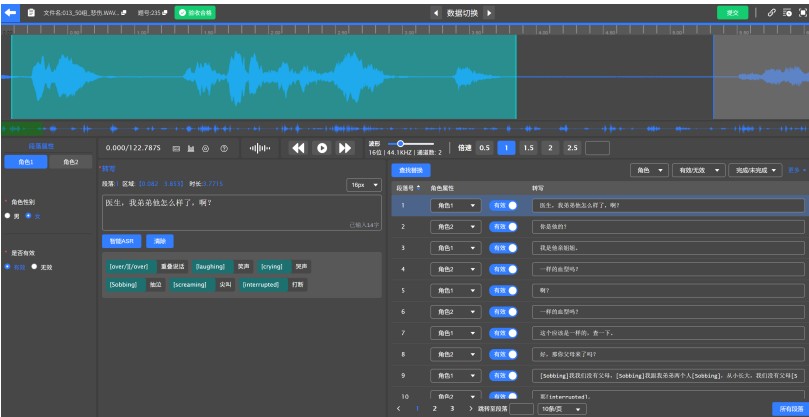

(a) Annotation platform of the speech emotion recognition.

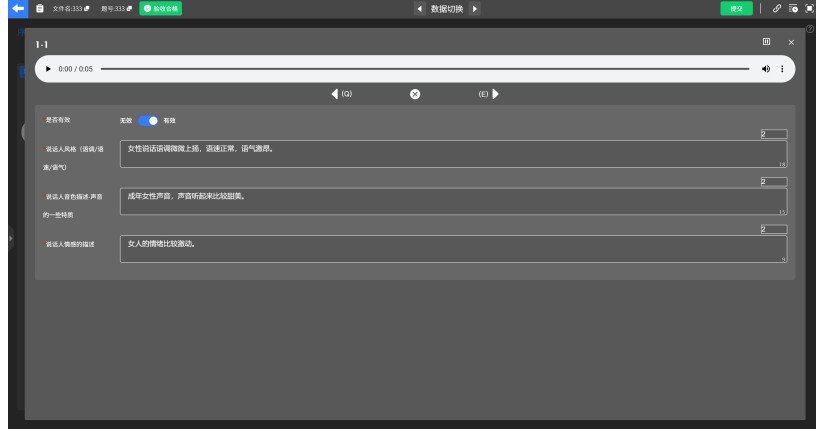

(b) Annotation platform of the emotional speaking style captioning.

Figure 3: Overview of the annotation platform interface.

(speech+text) does not yield substantial improvements over speech alone. The BiGRU model achieves 65.70% when both modalities are used, only slightly better than its speech-only counterpart. Similarly, DEP and EAMT see marginal, suggesting that the textual signal may offer limited complementary information for EPC.

In Emotion Recognition in Conversation (ERC), a similar modality gap is evident, with speech-based models outperforming their text-based versions by a large margin Hazarika et al. (2018); Yeh et al. (2019); Majumder et al. (2019); Ghosal et al. (2019), as shown in Table 13. DialoguGCN Ghosal et al. (2019) achieves the best performance among all models with 67.51% accuracy, leveraging its graph-based structure to effectively capture speaker interactions and contextual flow. DialogueRNN Majumder et al. (2019) and CMN Hazarika et al. (2018) also perform competitively in the speech modality (66.34% and 66.37%, respectively), while ICON Yeh et al. (2019) slightly lags behind at 65.31%. In contrast, their text-based counterparts yield significantly lower results—DialoguGCN at 49.75% and DialogueRNN at 49.63%—underscoring the limitations of relying solely on lexical information for emotion recognition. Notably, DialoguGCN consistently outperforms the other models across both modalities, suggesting its architectural advantage in handling complex conversational dynamics.

All models are trained using the Adam optimizer with an initial learning rate of 1e-4 and a weight decay of 1e-5. The batch size is set to 16, and models are trained for a maximum of 30 epochs (except for DialogueRNN, which is trained for 100 epochs). A StepLR scheduler is employed to decay the learning rate by a factor of 0.1 every 10 epochs in ERC.

## E  LIMITATIONS

The EmotionTalk dataset serves as an important resource in the field of conversational emotion recognition, providing a valuable experimental foundation for related research. However, when conducting in-depth analysis, it is necessary to examine several of its characteristics in order to more comprehensively evaluate its applicability across different research scenarios. First, it is worth noting that this dataset has a relatively limited scale, containing only 19 participants and 23.6 hours of multimodal data. Although the dataset demonstrates excellence in ensuring data quality and annotation precision, the limited sample size may to some extent affect the generalization ability of models trained on this dataset when applied to broader populations and diverse conversational scenarios.

These characteristics are not fundamental flaws of the dataset, but rather products of its specific research design, pointing researchers toward future development directions. For example, in future research, one could consider using the EmotionTalk dataset as an initial validation set and combining it with other larger-scale or more robust datasets to construct conversational emotion recognition systems with greater robustness and generalizability.

## F  LLMS USAGE

In the writing process of this study, large language models (LLMs) are employed as auxiliary tools, primarily for optimizing linguistic expression and standardizing formatting. These tools assisted in improving the clarity of presentation, logical coherence, and linguistic accuracy of the manuscript. It should be emphasized that the core concepts, experimental design, and conclusions of this research represent entirely original work by the authors. The authors assume full responsibility for the academic integrity of the research content, strictly adhering to academic ethical standards and ensuring the originality and authenticity of the research findings.

