# OpenReview forum: "EmotionTalk: An Interactive Chinese Multimodal Emotion Dataset With Rich Annotations"
_ICLR.cc/2026/Conference — ICLR 2026 Conference Withdrawn Submission_

### Official Review · Reviewer_ypS2 · 2025-10-29

**Soundness:** 2
**Presentation:** 3
**Contribution:** 3
**Rating:** 4
**Confidence:** 4

**Summary:**

The paper introduces EmotionTalk, a large-scale, high-quality Chinese multimodal emotion dataset that integrates text, audio, and video modalities. The dataset contains 23.6 hours of recordings from 19 professional actors engaged in 744 dyadic dialogues (19,250 utterances).

The authors emphasize data authenticity (improvised dialogues, not scripts), annotation rigor (five annotators + confidence weighting + negotiation rounds), and broad applicability (emotion recognition, sentiment analysis, emotion captioning).

The proposed dataset addresses the severe lack of large-scale Chinese multimodal datasets for emotion analysis.  Ensures authentic emotional expression via semi-improvised dialogue performed by trained actors.

However, the work mainly contributes a dataset; experimental analysis (fusion and recognition models) is mostly confirmatory and relies on existing architectures.

**Strengths:**

1. High-quality and novel resource: it addresses the severe lack of large-scale Chinese multimodal datasets for emotion analysis.

2. Rich and well-structured annotation scheme: it contains multi-level labeling (discrete, continuous, and descriptive captions) enables diverse tasks beyond classification.

3. Comprehensive experiments and baselines: it evaluates over 20 models (speech, vision, and text) and multiple fusion methods (TFN, MISA, LMF, etc.).

**Weaknesses:**

1. Overemphasis on dataset construction, limited methodological novelty: the work mainly contributes a dataset; experimental analysis (fusion and recognition models) is mostly confirmatory and relies on existing architectures.

2. Lack of comparison to other Chinese datasets: while CH-SIMS, M3ED, and MC-EIUch are mentioned, quantitative comparisons (data quality, annotation agreement, or inter-modal correlation) are missing.

3. Emotion captioning evaluation lacks human judgment: The automatic metrics (BLEU, ROUGE, BERTScore) may not adequately reflect caption quality or emotional appropriateness. A small-scale human evaluation would strengthen credibility.

**Questions:**

See weaknesses.

---

### Official Review · Reviewer_QfPS · 2025-10-31

**Soundness:** 2
**Presentation:** 3
**Contribution:** 2
**Rating:** 2
**Confidence:** 4

**Summary:**

This paper introduces EmotionTalk, the first high-quality Chinese multimodal emotion dataset designed for emotion recognition research. It features 23.6 hours of dyadic conversations (19,250 utterances) from 19 actors, annotated with 7 emotion categories, 5 sentiment levels, and 4 speech caption dimensions across acoustic, visual, and textual modalities. EmotionTalk addresses the scarcity of Chinese multimodal dialogue datasets and enables studies on unimodal and multimodal emotion recognition, missing modality challenges, and speech captioning tasks. Experiments validate its effectiveness, and the dataset will be freely available for academic use.

**Strengths:**

The results of the experiment are promising.

**Weaknesses:**

1. The authors emphasize the dataset's novelty by highlighting its high-quality recordings conducted by professional actors, improvisational dialogue design, and comprehensive multimodal annotations across text, audio, and video. However, the paper could further clarify how the novelty of EmotionTalk compares to existing datasets beyond general claims. For instance, while the improvisational approach is presented as a novel methodology to enhance emotional authenticity, the paper lacks quantitative or qualitative evidence proving that this approach results in superior realism compared to datasets like IEMOCAP or CH-SIMS. Moreover, the paper could benefit from a more detailed discussion of how the multimodal annotation system (e.g., the four fine-grained emotional speaking style captions) contributes uniquely to affective computing tasks compared to simpler annotation frameworks.
2. Although the paper's main focus lies in the dataset creation and experimental validation, there is limited theoretical analysis of the underlying framework or proposed methodologies. For instance, the inclusion of fine-grained emotional speaking style captions is described as innovative, but the paper does not provide a theoretical framework or empirical analysis justifying its relevance to downstream tasks. How do these captions improve model interpretability, generalization, or performance compared to using only traditional emotion labels?
3. The proposed weighted confidence score (Equation 1) and multi-round annotation negotiation process are interesting contributions, but the paper does not analyze how these processes impact annotation consistency, inter-rater agreement, or model accuracy.

**Questions:**

See Weaknesses.

---

### Official Review · Reviewer_T25u · 2025-10-31

**Soundness:** 2
**Presentation:** 3
**Contribution:** 2
**Rating:** 2
**Confidence:** 5

**Summary:**

The paper introduces EmotionTalk, a new Chinese multimodal emotion corpus comprising 23.6 hours of dyadic conversational data from 19 professional actors. The corpus provides annotations across three modalities (audio, video and text) and includes seven discrete emotion labels, five-dimensional sentiment intensity scores and novel 'emotional speaking style captions' across four dimensions (speaker, style, emotion and overall). The authors benchmark the corpus against unimodal and multimodal emotion recognition and captioning tasks, employing a variety of encoders and fusion methods.

**Strengths:**

1) The use of professional actors, theme-driven improvisation and a controlled recording environment produces more naturalistic expressions than scraping TV/movies.
2) Separate modality-specific labels, confidence scores, and Fleiss' Kappa scores demonstrate the seriousness of the annotation effort across modalities.
3) The evaluation of encoders and fusion methods provides a useful point of reference for future works.

**Weaknesses:**

1) DeepSeek-R1 generates emotional captions for speech and presents them as authoritative annotations. However, the process of human verification is unclear, as are the methods used to agree on the quality of captions, verify semantic accuracy, and research preferences. This raises a significant risk of LLM hallucinations or stylistic artifacts appearing in the corpus.
2) Despite the severe class imbalance, the evaluation relies heavily on accuracy.
3) Although the paper highlights cross-modal label disagreement as a key motivation, it does not analyze whether multimodal models actually outperform unimodal ones in these specific cases.
4) The corpus comprises only 19 professional actors, who are likely to come from similar socioeconomic and cultural backgrounds. This raises concerns about its applicability to broader populations, particularly in light of the recognized cultural and individual variations in emotional expression.
5) Although the 'emotional speaking style caption' is presented as innovative, it is not clearly differentiated from existing constructs such as paralinguistic descriptors, prosodic tags or style prompts. While the four dimensions (speaker, style, emotion and overall) are listed, they are not formally defined or validated against linguistic or psychological frameworks.
6) The paper acknowledges that different modalities yield different labels, but does not explore whether these differences reflect genuine perceptual distinctions or artifacts of the annotation protocol.
7) Although the paper uses DeepSeek-R1 to generate five variants of each caption, it provides no analysis of their semantic diversity, redundancy or utility. Are all five necessary? Do they improve model training? These questions remain unanswered.
8) The paper randomly splits the data at the utterance level (8:1:1), but does not ensure that the splits are speaker-disjoint. This increases the risk of data leakage, as the same speaker appears in both the training and test sets. This can inflate performance estimates due to speaker identity cues rather than an understanding of emotion.
9) It is surprising that the paper does not evaluate zero-shot emotion recognition using LLMs with prompting (in-context learning), given the use of LLMs for caption generation. This omission undermines the assertion that fine-tuning on EmotionTalk is essential or advantageous.

**Questions:**

1) Were the signatures generated by the LLM verified by humans? If so, how was this done? If not, why are they considered a reliable form of markup?
2) Why aren't other metrics that more accurately account for the strong class imbalance included in the experiments?
3) Have you analyzed the performance of the model specifically using examples with cross-modal inconsistency?
4) Why is the division into training, validation and test sets not speaker-independent? Does this not result in information leakage and an overestimation of metrics?
5) How representative of the population are the 19 actors? Is there any data available on their age, gender or place of birth?
6) How did you assess the 'naturalness' of improvised dialogue compared to real-life conversations? Are there any objective metrics?
7) How did you determine that the five LLM signature variants were genuinely diverse and useful? Was an analysis of their semantic variation conducted?
8) Why haven't the best LLMs been tested using zero-shot methods for emotion recognition without fine-tuning?
9) Are there any examples of speech overlay?

---

### Note · Authors · 2025-11-21

I have read and agree with the venue's withdrawal policy on behalf of myself and my co-authors.